# Rabies Virus Populations in Humans and Mice Show Minor Inter-Host Variability within Various Central Nervous System Regions and Peripheral Tissues

**DOI:** 10.3390/v14122661

**Published:** 2022-11-28

**Authors:** Carmen W. E. Embregts, Elmoubashar A. B. A. Farag, Devendra Bansal, Marjan Boter, Anne van der Linden, Vincent P. Vaes, Ingeborg van Middelkoop-van den Berg, Jeroen. IJpelaar, Hisham Ziglam, Peter V. Coyle, Imad Ibrahim, Khaled A. Mohran, Muneera Mohammed Saleh Alrajhi, Md. Mazharul Islam, Randa Abdeen, Abdul Aziz Al-Zeyara, Nidal Mahmoud Younis, Hamad Eid Al-Romaihi, Mohammad Hamad J. AlThani, Reina S. Sikkema, Marion P. G. Koopmans, Bas B. Oude Munnink, Corine H. GeurtsvanKessel

**Affiliations:** 1Department of Viroscience, Erasmus Medical Centre, 3015 GD Rotterdam, The Netherlands; 2Ministry of Public Health, Doha P.O. Box 42, Qatar; 3Hamad Medical Corporation, Doha P.O. Box 3050, Qatar; 4Biomedical Research Centre, Qatar University, Doha P.O. Box 2713, Qatar; 5Department of Animal Resources, Ministry of Municipality, Doha P.O. Box 35081, Qatar; 6Biotechnology Departments ERC, Animal Health Research Institute, Dokki 12611, Egypt

**Keywords:** rabies virus, NGS, minor variants, CNS

## Abstract

Rabies virus (RABV) has a broad host range and infects multiple cell types throughout the infection cycle. Next-generation sequencing (NGS) and minor variant analysis are powerful tools for studying virus populations within specific hosts and tissues, leading to novel insights into the mechanisms of host-switching and key factors for infecting specific cell types. In this study we investigated RABV populations and minor variants in both original (non-passaged) samples and in vitro-passaged isolates of various CNS regions (hippocampus, medulla oblongata and spinal cord) of a fatal human rabies case, and of multiple CNS and non-CNS tissues of experimentally infected mice. No differences in virus populations were detected between the human CNS regions, and only one non-synonymous single nucleotide polymorphism (SNP) was detected in the fifth in vitro passage of virus isolated from the spinal cord. However, the appearance of this SNP shows the importance of sequencing newly passaged virus stocks before further use. Similarly, we did not detect apparent differences in virus populations isolated from different CNS and non-CNS tissues of experimentally infected mice. Sequencing of viruses obtained from pharyngeal swab and salivary gland proved difficult, and we propose methods for improving sampling.

## 1. Introduction

Rabies is a viral encephalitis that is caused by viruses of the genus *Lyssavirus*, and an estimated 99% of human rabies cases are caused by rabies virus (RABV) transmitted by dogs [1]. Lyssaviruses are transmitted through the saliva of lyssavirus-infected animals, and rapid post-exposure prophylaxis is essential to prevent development of the disease. There are, however, no treatment options after the onset of neurological symptoms, which makes rabies the deadliest zoonosis worldwide with a near 100% case-fatality rate. Every year at least 59,000 human rabies fatalities are reported worldwide, but the true burden is expected to be much higher than that due to diagnostic difficulties, especially in rabies-endemic regions [2,3]. Furthermore, loss of livestock (cattle, sheep, goats) due to RABV infections is a major problem in South America [4].

Lyssaviruses are neurotropic and can infect peripheral nerve endings, after which they travel to the spinal cord via the ventral or dorsal root ganglions. After reaching the brain and infecting local neurons, the virus further spreads to the salivary gland and other organs and is finally secreted through the saliva. Despite their neurotrophic nature, lyssaviruses are capable of infecting muscle cells at the initial infection site, various cell types during the centrifugal spread, and acinar cells in the salivary gland, of which the latter step is essential for virus secretion and infection of the next host [5,6].

RABV, along with other lyssaviruses, are negative single-stranded RNA viruses (-ssRNA). These RNA viruses are generally known to display high rates of evolutionary change by both natural selection and recombination, enabling them to rapidly adapt to new hosts and cell types [7,8,9]. Interestingly, RABV shows a rather low mutation rate when compared to other -ssRNA viruses [10]. While most lyssaviruses are thought to be restricted to specific hosts, RABV is transmitted by multiple species of the orders Carnivora and Chiroptera (bats) [11]. In addition, rare spill-over infections have been documented in a variety of other mammals, including camels and kudus [12,13]. Studying adaptability of RABV within different environments is of immense importance, given its ability to thrive in wide range of hosts by infecting different types of cells throughout its infection cycle [14,15,16]. Comparing virus populations in different organs, both inside and outside of the CNS during different phases of infection, is a first step in understanding the genetic plasticity of RABV.

While most viral sequencing is performed on brain tissue specimens, given the presence of high viral loads, there is less information about viral populations in other organs. In this study we investigated RABV populations in three different central nervous system (CNS) regions (hippocampus, medulla oblongata and spinal cord) of a fatal human rabies case [12], and the adaptation of the isolated virus after in vitro passages in mouse neuroblastoma cells. In parallel, we investigated RABV populations in various CNS (brain, trigeminal ganglion, dorsal and ventral spinal cord) and non-CNS (salivary gland, tongue, nuchal skin and pharyngeal swab) biopsies of C57BL/6 mice that were experimentally infected with a silver-haired bat rabies virus (SHBRV) reference strain [17,18]. By using next-generation sequencing (NGS) and minor variant analysis, we were able to describe the virus population within various CNS biopsies of a fatal human RABV case, as well as the virus population within various CNS and non-CNS tissue biopsies of mice infected with a bat-related RABV strain, in great detail. While minor variants were found to play an insignificant role in RABV host switch events [19], investigating minor variants may provide crucial insights into where mutations take place, and which variants might become dominant.

Pharyngeal swabs contain secreted virus and are therefore an important specimen for studying virus evolution. However, the low virus loads obtained from pharyngeal swabs severely limit the possibility of investigating virus populations. The parotid salivary gland is easy to sample and contains saliva that is ready to be secreted. Therefore, we also assessed the suitability of using salivary gland biopsies as a read-out for secreted virus.

## 2. Materials and Methods

### 2.1. Human Materials and Cell Culture

Human CNS biopsies (medulla oblongata, hippocampus, spinal cord) were obtained from a fatal human rabies case in Qatar described previously [12]. Samples were stored in virus transport medium and stored at −80 °C until processing. Samples were homogenized in DMEM (Lonza), centrifuged for five minutes at 5000× *g*, after which 200 µL of the supernatant was filtered using a 0.45 µm filter and was incubated on mouse neuroblastoma astrocytes (MNA) for 1 h. After removal of the inoculum the cells were incubated with supplemented DMEM medium at 37 °C in the presence of 5% CO_2_. Virus was passaged when cytopathic effect was observed; the cell culture supernatant was centrifuged and filtered, and 200 µL was inoculated onto freshly seeded MNA cells. The remainder of the filtered virus was stored at −80 °C.

### 2.2. Animal Materials

Six-to-eight-week-old mice (C57BL/6) were intramuscularly inoculated with 10^5^ or 10^6^ TCID_50_ of silver-haired bat rabies virus (SHBRV) in the left hind leg. This experiment was designed and performed to validate an in vivo infection model, and the samples used in this study were taken after the initial experiment had finished. Mice were euthanized upon showing neurological symptoms, which appeared between day six and eight. Biopsies of target organs and tissues of interest (brain, trigeminal ganglion, spinal cord dorsal and ventral horn, parotid salivary gland and tongue), as well as a pharyngeal swab were taken, homogenized in 1 mL of virus-transport medium, and stored at −80 °C until further processing. In parallel, organs were fixed in 10% neutral-buffered formalin to verify virus protein expression by immunohistochemistry (IHC). Samples were taken from 10 mice in total, from which the samples of four animals were used to investigate differences in viral populations within the salivary gland and pharyngeal swab isolates, and the samples of the remaining six animals were used to compare virus populations throughout different organs (brain, trigeminal ganglion, spinal cord dorsal and ventral horn, tongue, nuchal skin, and parotid salivary gland). All animal experiments were performed in compliance with Dutch legislation for the protection of animals used for scientific purposes (implementing EU Directive 2010/63) and other relevant regulations. The research was conducted under a project license (AVD1010020187204) approved by the competent Dutch authority. The specific study protocol (18−7204−01) was approved by the institutional Animal Welfare Body.

### 2.3. Sample Preparation and RNA Isolation

Isolates from original materials from the human RABV patient, the third and fifth passage of these materials on MNA cells, and the samples taken from the infected mice, were included for RNA isolation and sequencing. All samples were centrifuged and filtered through a 0.45 µm filter, after which 100 µL of the filtrate was mixed with lysis buffer (Qiagen). The animal samples were pre-processed with Omnicleave (Lucigen) to cleave present host RNA, and all samples were further processed using the High Pure RNA isolation kit (Roche) following the manufacturers guidelines, including the on-column DNA digestion step. RT-PCR was performed as an initial investigation on the presence of viral RNA, using a RABV genotype 1 RT-PCR [20] on all isolates obtained from mice tissues, and a pan-lyssavirus RT-PCR [21] on the isolates and passages obtained from the human materials.

### 2.4. cDNA Library Preparation and Sequencing

cDNA was made using random primers (Thermo Fisher) and SuperScript IV (Thermo Fisher) and dsDNA was made using Klenow (NEBNext). The KAPA Hyper Plus kit (Roche) was used to prepare the library for sequencing with minor adjustments: fragmentation time was reduced to 3 min and the adapters were diluted 1:10. Targeted enrichment was performed using VirCapSeq [22] and all 50 samples were pooled equimolarly and sequenced on an Illumina MiSeq v3 flow cell (2 × 300 bp).

### 2.5. Minor Variant Analysis

Consensus sequences were generated using a reference-based alignment against the previously sequenced SHBRV-18 strain (AY705373.1) or the RABV_Nepal_2018 strain (MN534894.1). The reads were re-aligned to the newly generated consensus sequences and minor variants with a 20% frequency cutoff were determined using Geneious version 9.1.8 using default settings.

### 2.6. Immunohistochemistry and Immunofluorescence

Presence of viral proteins in formalin-fixed paraffin-embedded tissue biopsies was verified by immunohistochemistry (IHC), and presence of virus in the cell culture passages was verified by immunofluorescence (IF). Briefly, 3 µM slides were cut from paraffin-embedded tissues of both human and murine samples, and the RABV nucleoprotein (RABV-N) was detected using the 5DF12 antibody (kindly provided by P. Koraka). Images were acquired using an Olympus BX51 microscope. MNA cultures were fixed with ice-cold 80% acetone and were stained with the FITC-conjugated anti-RABV-N antibody (Fujirebio). A nuclear counterstain with Evans blue was included and images were acquired using a Zeiss AX10 Colibri 7 fluorescence microscope.

## 3. Results

NGS of RABV isolated from three regions of the CNS of a fatal human case (medulla oblongata, hippocampus and spinal cord), as well as the following in vitro passages in mouse neuroblastoma (MNA) cells (illustrated in Figure 1), was successful. No single nucleotide polymorphisms (SNP) were detected in the consensus sequence or a minor variant of original materials of the three different tissues, nor in the first in vitro passage (Table 1). One non-synonymous SNP within the matrix (M2) gene was first observed in the third in vitro passage of the spinal cord which became dominant in the fifth passage. Different dominant SNPs were found in the third and fifth passage of the virus isolated from medulla oblongata and the fifth passage of the virus isolated from the hippocampus, but these did not lead to codon changes. One minor variant was detected in the third in vitro passage of virus isolated from the spinal cord, but it decreased below the 20% threshold in the fifth in vitro passage. No minor variants were detected throughout the passages of the hippocampus, and only in passage three, three minor variants were detected in virus isolated from the medulla oblongata.

To investigate the possible occurrence of organ-specific lyssavirus adaptations, we sequenced virus isolated from different CNS tissues (brain, trigeminal ganglion, spinal cord) and non-CNS tissues (nuchal skin, tongue epithelium, salivary gland and pharyngeal swab) of C57BL/6 mice that were experimentally infected with the silver-haired bat rabies virus strain (SHBRV, Figure 2A), a strain for which the disease progression has been well described before in our lab [17,18]. In contrast to the abundant presence of viral proteins in CNS tissues, only very few RABV-positive cells were found in non-CNS tissues (Figure 2B), which is reflected in the higher Ct values of non-CNS tissue biopsies (Figure 2C and Appendix A).

RABV is predominantly secreted via saliva, making saliva the preferred sample for investigating secreted RABV populations. However, obtaining enough saliva sample to successfully culture or sequence secreted virus is often a challenge, especially when working with small experimental animals. We therefore investigated if the salivary gland tissue is a reliable source for characterizing secreted virus populations by performing pairwise collection and sequencing of viruses isolated from pharyngeal swabs and salivary gland biopsies obtained from mice 1–4. Despite the presence of RABV as detected by PCR, only one of these pairwise comparisons (mouse 3) resulted in successful viral sequences of both samples. In this pairwise sample comparison, we found one SNP and three minor variants in the salivary gland isolate and two different SNPs in the pharyngeal swab isolate, of which one resulted in an amino acid change in the polymerase gene (Table 2). The second sequence obtained from a pharyngeal swab isolate contained one non-synonymous SNP in the matrix protein, a mutation that was not observed in any other sample. No other samples were sequenced from mice 1–4, and therefore we cannot conclude if the observed mutations are organ-specific. Moreover, no specific SNPs were detected in the salivary gland isolates of mice eight and nine.

To compare virus populations in CNS tissues with non CNS tissues, isolates from both CNS (brain, trigeminal ganglion, spinal cord ventral and dorsal) and non-CNS (tongue epithelium, nuchal skin, salivary gland) tissues from mice 5–10 were collected and used for sequencing. Successful sequencing required a Ct value below ~28–29 (Figure 2C and Appendix A). Due to low viral loads in the non-CNS tissues, only the isolates obtained from tissues of mice eight and nine resulted in successful sequencing of all seven different tissues. No major differences were detected between samples of the individual mice. The only SNP detected in virus isolated from the trigeminal ganglion and tongue epithelial was present as a minor variant in all tissue isolates except the salivary gland of mouse eight. Additionally, all tissue isolates from mouse nine, but none of the other samples included in the experiment, contained the same SNPs. The pattern that SNPs in our dataset group together on the individual host (mouse) level, and not on organ level, was strengthened by the fact that only one SNP was observed in more than one mice. A non-synonymous SNP in the RABV glycoprotein protein gene was observed in the nuchal skin sample from mouse nine.

A limited number of minor variants were detected in this dataset, similar to the SNPs. However, the majority of them were found in single animals and did not appear to be tissue-specific, as they were observed as SNP in other isolates of the same mouse.

## 4. Discussion

In the presented study we investigated virus populations isolated from human CNS tissues, and from several CNS- and non-CNS tissues from experimentally infected mice. Although we did not detect RABV evolution within the analyzed human CNS specimen, multiple SNPs and minor variants were detected within few in vitro passages. Rapid virus adaptation to in vitro culture has been described for RABV previously [14,15,23], and mutations were observed both during homologous and heterologous culture conditions [14]. The observed changes might be indicative of either cell-culture or host adaptations, given that the in vitro passages were performed on mouse neuroblastoma cells and not on cells of human origin. Irrespective of the nature of the mutations, their presence indicates the need for using low-passage stocks for infection experiments, as well as the necessity of sequencing every in vitro passage before using them in experiments.

While the Ct levels in the salivary gland extracts were lower than in pharyngeal swabs of mice, only one of the pairwise comparisons between salivary gland and pharyngeal swab isolates resulted in successful viral sequences of both samples. Given the low number of successful sequences obtained from salivary gland isolates, we cannot definitively conclude if the salivary gland is a reliable source for studying secreted lyssaviruses. Most likely the high level of host RNA in salivary glands resulted in this failure to retrieve complete viral sequences. Furthermore, high levels of RNAses present in the parotid salivary gland are known to drastically reduce the yield of high-quality RNA isolated from its tissue [24]. Given the simplicity of collecting a pharyngeal swab, we propose an optimization of the sample collection protocol to allow for successful sequencing in future experiments. This includes lowering the volume of virus-transport medium in order to reduce dilution of the sample, storing the swab directly in lysis buffer, and using a swab of different materials, since foam swabs were found to be the optimal choice for virus collection [25,26]. The presence of these minor variants in the salivary gland, but not in the pharyngeal swab, might be explained by the presence of virus originating from (para-)sympathetic nerves that innervate the salivary gland [27].

In line with a previous study [14], no differences were found in virus populations isolated from CNS and salivary gland biopsies. Besides the salivary gland, no differences were detected in virus population isolated from other non-CNS tissues, indicating that RABV does not require adaptations in order to spread between the different tissues and cell types of its host. While the virus can infect various extra-neuronal tissues and cells of non-neuronal origin, the secreted virus needs to be capable of infecting (peripheral) nerves again. Therefore, large organ-specific adaptations are not to be expected, since this would not be beneficial to the spread and infectivity of the virus. Throughout the dataset we detected one SNP in the RABV glycoprotein, an essential protein for binding to host receptors. Changes in the glycoprotein gene sequence are commonly associated with changes in virulence and/or adaptations to a novel host [28,29,30,31]. However, this SNP was detected in virus isolated from nuchal skin, a tissue that is normally not actively involved in virus dissemination.

In conclusion, this is the first study that performed NGS and minor variant analysis on virus isolated from a broad range of CNS and non-CNS samples of human and mouse origin. The mice experiment was performed independently from the collection and in vitro passaging of the human CNS isolates. While this resulted in insights of inter-host adaptations of both a dog-related RABV strain (human case) and a bat-related RABV strain (mice experiment), more in-depth insights into the dog-related strain could be obtained by performing a specifically designed animal experiment that uses a dog-related RABV isolate. Altogether, better understanding of the evolution and adaptation of RABV in the host will be valuable in increasing the understanding of the spread of RABV and related lyssaviruses within different hosts.

## Figures and Tables

**Figure 1 viruses-14-02661-f001:**
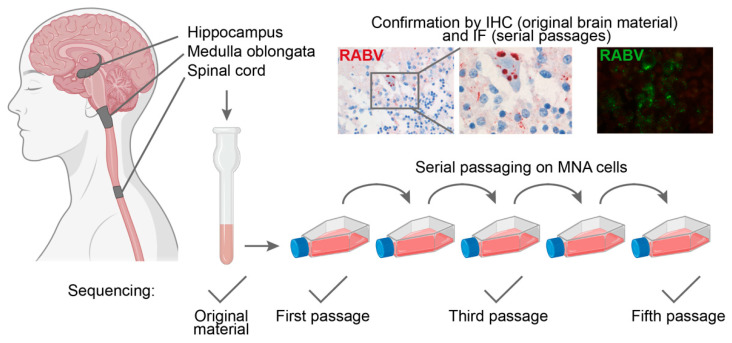
Overview of the human central nervous system (CNS) biopsy isolates used for in vitro passaging and sequencing. The grey areas in the CNS depicted on the left indicate the three areas that were included in our study. The expression of viral proteins within these samples was verified with immunohistochemistry (IHC, red staining). Serial passaging of the isolates was performed on mouse neuroblastoma cells (MNA), and the first, third and fifth passage was used for sequencing. Presence of virus within the passages was confirmed by PCR and immunofluorescence (IF, green staining top right picture). The figure, except the IHC and IF pictures, was generated using BioRender.com.

**Figure 2 viruses-14-02661-f002:**
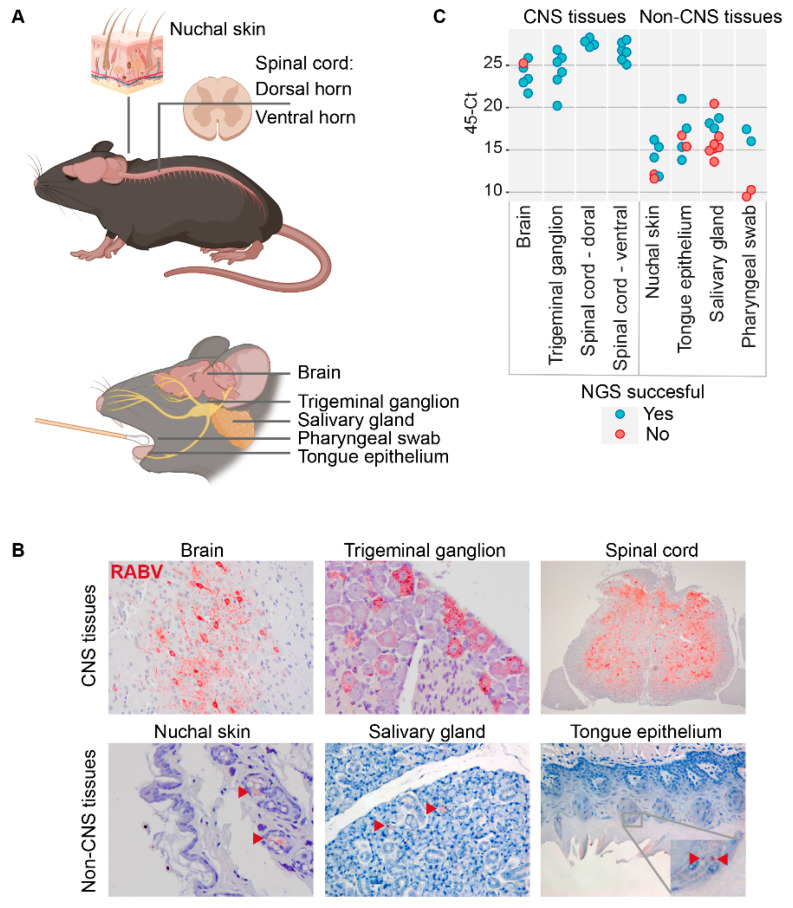
Overview of the mice tissue isolates used for sequencing, and verification of the infection by IHC and PCR. All samples processed for PCR and subsequent sequencing are shown in (**A**). Presence of viral proteins was verified by IHC (red staining) (**B**). Red arrows in the lower panel show the sparse distribution of RABV-positive cells in non-CNS tissues. The table in (**C**) presents the 45-Ct values indicating the viral loads of each specified organ. Green cells resulted in a successful sequencing result, grey cells yielded incomplete sequencing runs and were therefore excluded for further analysis. Panel (**A**) was generated using BioRender.com.

**Table 1 viruses-14-02661-t001:** Overview of the SNPs and minor variants found in RABV isolated from different human CNS regions (original material, OM) and in vitro passages (P1, P3, P5) of these materials. The presence and corresponding percentages of the specific minor variants are indicated in blue; orange indicates that the specific minor variant is presented as a SNP in another tissue. Grey indicates an ongoing nucleotide shift (49.6%C, 47.4%T).

Passage	Tissue	SNPs (>50%)	Minor Variants (>20%)				
			A2397G	A2413C	T2699C	T2977Y	T3445G
OM	Medulla oblongata						
	Spinal cord						
	Hippocampus						
P1	Medulla oblongata						
	Spinal cord						
	Hippocampus						
P3	Medulla oblongata						
	Spinal cord	T2977Y (M2: S185P)		21.6			
	Hippocampus	A2399G (non-coding)					
P5	Medulla oblongata	G2397A (non-coding)	43.4		20.8		20.1
	Spinal cord	T2977C (M2: S185P)				>50	
	Hippocampus	A2399G (non-coding)					

**Table 2 viruses-14-02661-t002:** Overview of SNPs and minor variants in the sequenced mice tissues. Samples are ordered per mice, the SNPs (>50% abundance) are included in the third column, with the indicated amino acid change (if present) in bold. Minor variants, with a cutoff of 20%, are shown from column 3 onwards. The presences of the specific minor variants are indicated in blue, orange indicates that the specific minor variants are presented as a SNP in another tissue. An overview of the abundance (%) of the minor variants can be found in Appendix A.

Mouse ID	Organ	SNPs (>50%)	Minor Variants (>20%)															
			A247C	C2406A	T4600C (G:R456W)	A5315G (L:G32D)	T5898G (L:D226E)	A6771G	A7704G	G7704A	A8313G	T10668C	C10668T	G10785A	C11031T	T11031C	A11775G	G11775A
2	Pharyngeal swab	A1617C (M1:L62T), C10668T																
3	Salivary gland	C1783T																
3	Pharyngeal swab	C1702T, T5898G (L:D226E)																
5	Trigeminal ganglion	G3131T																
5	Spinal cord—dorsal	G3131T																
5	Spinal cord—ventral	G3131T																
6	Brain	C247A																
6	Trigeminal ganglion	C247A																
6	Spinal cord—dorsal	C247A																
6	Spinal cord—ventral	C247A																
6	Nuchal skin	C247A, C2552T																
7	Brain	A2406C, C5246T																
7	Trigeminal ganglion	A2406C, C5246T																
7	Spinal cord—dorsal	A2406C, C5246T																
7	Spinal cord—ventral	A2406C, C5246T																
7	Nuchal skin	A2406C																
8	Brain																	
8	Trigeminal ganglion	C10668T																
8	Spinal cord—dorsal																	
8	Spinal cord—ventral																	
8	Nuchal skin																	
8	Tongue epithelium	C10668T																
8	Salivary gland																	
9	Brain																	
9	Trigeminal ganglion																	
9	Spinal cord—dorsal																	
9	Spinal cord—ventral																	
9	Nuchal skin	T4600C (G:R456W)																
9	Tongue epithelium																	
9	Salivary gland																	
10	Brain																	
10	Trigeminal ganglion																	
10	Spinal cord—dorsal	A7704G, C11031T, G11775A																

## Data Availability

The data presented in this study are available on request from the corresponding author.

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
