# Peer review of "Rabies Virus Populations in Humans and Mice Show Minor Inter-Host Variability within Various Central Nervous System Regions and Peripheral Tissues"

_viruses, 2022, doi:10.3390/v14122661_

Round 1

Reviewer 1 Report

The authors have used human biopsy material, cell passages thereof and samples from mouse inoculation experiments to identify RABV populations and minor variants. Using state-of-the-art techniques, the authors found very little differences in virus populations.

While technically sound I have some reservations with this study. First, the hypothesis/statement that RABV displays, similar to other negative single-stranded RNA viruses displays high rates of evolutionary change is not substantiated. In fact, the contrary is the case with RABV showing a rather low mutation rate (PMID: 18585530). It is correct that the genetic diversity (based on consensus sequence) within RABV also influences the likelihood of spillovers or even establishments in new hosts, but minor virus populations are not considered to play a role (PMID: 29255631). The entire study design appears rather fragmented, with samples being already available from human cases or animals trial sequenced and analysed in the hope of discovering something novel. The results (few SNPs and minor variants are present, and more likely after passaging) are not surprising and are somewhat to be expected.

Author Response

Dear Editor and Reviewers,

Your time and expertise for reviewing our manuscript “Rabies virus populations in human and mice show minor inter-host variability within various central nervous system regions and peripheral tissues. “ is highly appreciated. Please find below our point-to-point rebuttal, we hope that we answered your questions in a satisfactory manner, and that we improved the quality of our manuscript substantially.

Response to reviewer 1:

  1. The hypothesis/statement that RABV displays, similar to other negative single-stranded RNA viruses displays high rates of evolutionary change is not substantiated.

We made the requested changes to the introduction, and added a statement on the low evolutionary rate of RABV, including the suggested reference, in line 55-56. In line 77-80 we added a statement, and the suggested reference, on the insignificant role of minor variants in RABV host switch events.

  1. The entire study design appears rather fragmented, with samples being already available from human cases or animals trial sequenced and analyzed in the hope of discovering something novel.

We agree with the reviewer that the study layout is suboptimal, and we emphasize this in the discussion as well (line 287-291). Obtaining samples from human rabies patients is very difficult, given the dramatic course of disease and the invasive procedure that is required to obtain brain specimens. Due to ethical and religious regions, obtaining these samples is often not possible. The mice samples used in this study originated from another experiment. We used a “dual-use” approach where we used one animal study for multiple investigations, instead of performing additional animal studies, in an effort to reduce the total number of experimental animals. By sequencing both CNS and non-CNS samples from the experimentally infected mice we are still convinced that this study is of additive value to the scientific community. Furthermore, we propose future studies in the discussion, and we hope that the outcomes of this manuscript will inspire other groups to further investigate RABV virus populations as well.  

  1. Are all the cited references relevant to the research? Must be improved.

The suggested references have been added to the manuscript in line 56 and line 78.

Reviewer 2 Report

This is a novel approach to study rabies virus populations within specific hosts and tissues, i.e.  multiple CNS and non-CNS tissues of experimentally infected mice! The methodology and ethical clearance procedure have been well described. The results are consistent with virus adaptation to various tissues and organs in experimentally infected animal. Neurotropism of rabies virus is reiterated with experimental and clinical trials in CNS and non-CNS tissues of experimental animals. I agree with proposed methods for improving the sampling from pharyngeal swabs and salivary glands for detection of rabies viruses.  

Author Response

We want to thank the reviewer for the positive evaluation of the manuscript. The reviewer had no comments and suggestions for improving the manuscript.